# The Timing of Activity after Eating Affects the Glycaemic Response of Healthy Adults: A Randomised Controlled Trial

**DOI:** 10.3390/nu10111743

**Published:** 2018-11-13

**Authors:** Andrew N. Reynolds, Bernard J. Venn

**Affiliations:** 1Department of Human Nutrition, University of Otago, P.O. Box 56, Dunedin 9054, New Zealand; andrew.reynolds@otago.ac.nz; 2Edgar National Centre for Diabetes and Obesity Research, Dunedin School of Medicine, University of Otago, P.O. Box 56, Dunedin 9054, New Zealand

**Keywords:** postprandial, glycaemia, activity, exercise, timing

## Abstract

There is scant information on how a time lag between the cessation of eating and commencement of physical activity affects postprandial glycaemia. Starting at baseline (t = 0), participants ingested white bread containing 50 g of available carbohydrates within 10 min. Using two crossover conditions, we tested the effect over 2 h on postprandial glycaemia of participants undertaking light activity at 15 or 45 min following baseline and compared it with a sedentary control condition. The activity involved cycling on a stationary ergometer for 10 min at 40 revolutions per min with zero resistance. Seventy-eight healthy adults were randomized to the 15 or 45 min activity arm and then randomised to the order in which they undertook the active and sedentary conditions. Cycling 45 min after baseline changed the course of the blood glucose response (likelihood ratio chi square = 31.47, *p* < 0.01) and reduced mean blood glucose by 0.44 mmol/L (95% confidence interval 0.14 to 0.74) at 60 min when compared with the sedentary control. No differences in postprandial blood glucose response were observed when cycling started 15 min after baseline compared with the sedentary control. Undertaking activity after waiting for 30 min following eating might be optimal in modifying the glycaemic response.

## 1. Introduction

Poor blood glucose control is a risk factor for the development of type 2 diabetes [1] and cardiovascular disease [2,3,4], even when at subclinical levels [5,6]. Regular physical activity assists in maintaining blood glucose control [7,8], with activity-mediated skeletal muscle glucose uptake able to reduce circulating levels [9]. This is one reason that regular activity is widely promoted [10,11] to both the general population and subgroups of the population where internal glycaemic regulation may no longer be sufficient.

One aspect central to blood glucose control is the postprandial response. Repeated bouts of postprandial hyperglycaemia occurring over months and years result in accumulated micro- and macro-vascular damage [2,3,12], are the primary determinants of glycaemic variability [13] and are drivers of protein glycation [14]. It has been found that both pre- and postprandial physical activity in adults with normal glucose tolerance has dampened postprandial blood glucose excursions [15,16,17,18,19,20,21,22]. Based on the findings of a review, it has been concluded that activity commenced after eating produces a more favourable post-meal glycaemic response compared with a comparable amount of pre-meal activity [23]. A variable that has received little attention is the timing of commencement of activity following a meal. It has been suggested that the optimum timing for the commencement of post-meal activity is 30 min after finishing a meal using the rationale that this coincides with the greatest influx of dietary-derived glucose into the bloodstream [24]. Given the increase in glucose utilisation at higher concentrations of plasma glucose [25], it is feasible that activity undertaken at the blood glucose peak may be more effective at reducing blood glucose than when activity is taken during carbohydrate absorption.

However, the precision with which the timing of activity needs to be undertaken after eating is unclear. It has been found that light walking commenced immediately following a meal lowered postprandial glycaemia [26], as did activity commenced 30 min after the start of a meal [27]. In contrast, delaying the commencement of activity for one hour following the start of eating resulted in no glycaemic benefit compared with a sedentary condition [28]. However, within each of these studies there was no comparison of glycaemic effectiveness between activity started at different times after eating. Nor was there consistency in the duration or intensity of the activity. In the studies by Lunde et al. [26], Nelson et al. [27], and Borer et al. [28], the activities were slow walking for 20 min, cycle ergometer for 45 min at 55% VO_2_ max, and treadmill walking for two hours at 43% VO_2_ max respectively.

Effects of low-intensity activity carried out for a short duration after eating had variable effects on postprandial glucose excursions. Among 14 healthy women, slow walking for 15 min started immediately following a meal resulted in a 1.5 mmol/L reduction in blood glucose concentration at the end of the active period compared with a sedentary arm [22]. In contrast, blood glucose concentration was not different among 11 adults when eight minutes of moderate intensity cycling was undertaken immediately following eating compared with control [29] and was higher by ~1 mmol/L 30 min after finishing 15 min bouts of cycling by six healthy volunteers compared with a control arm [30]. There may be a number of reasons for the discrepant findings, including differences in participant demographics and study design, but of note, the sample numbers were small. Given this heterogeneity in findings we were interested in exploring whether low intensity activity over a short duration could influence postprandial glycaemia of a larger group. The duration and intensity of the activity are factors that require consideration if lowering postprandial glycaemia is a long-term goal requiring sustained adherence over years or a lifetime.

Thus, the primary aim of this experiment was to compare the effects on postprandial blood glucose concentration of undertaking activity at two timepoints commenced either 15 min (during glucose absorption) or 45 min (coinciding with peak glucose) after the consumption of a meal in comparison to a sedentary control. The activity chosen was a cycle ergometer set at zero resistance, in order that anyone could undertake the activity regardless of fitness, and for a duration of 10 min to ensure that in practice people would be more likely to have the time to commit to the activity after meals over months and years compared with a longer duration.

## 2. Materials and Methods

This randomised controlled trial was conducted between February and March of 2014 at the research clinic of the Department of Human Nutrition, University of Otago, Dunedin, New Zealand. This study has University of Otago Human Ethics Committee approval (09/012) and is registered with the Australian New Zealand Clinical Trials Registry (ACTRN12614000264684).

### 2.1. Study Design

The study was designed as two crossover trials run in parallel (Figure 1). A three-arm crossover would have been another option but, due to resourcing constraints, it was more efficient to undertake the experiment as described.

### 2.2. Participants

We recruited young adults without a self-reported diagnosis of dysglycaemia. Diagnosed diabetes mellitus, cardiovascular disease, cancer, diseases of the digestive system, food allergies, and pregnancy were exclusion criteria for study participation. The study was designed as a crossover (activity vs. sedentary), but also as a randomised parallel trial in which two groups were studied: Group 1, in which the activity arm was commenced 5 min after eating (15 min from baseline), and Group 2, in which the activity commenced 35 min after eating (45 min from baseline).

### 2.3. Randomisation

Allocation to group (15 or 45 min activity start) and allocation to order (active or sedentary) was achieved using a computer-generated randomisation protocol. Randomisation took place at a separate site before study commencement.

### 2.4. Intervention

Each participant attended two fasted morning tests. Participants were advised to avoid alcohol, caffeine, and be consistent with their physical activity level and diet in the 24 h before each morning test. Participants were advised to walk slowly or drive to the test facility each morning and were seated for a minimum of five minutes before testing commenced. Morning tests were separated by a one-week washout.

Each morning the participants consumed a weighed amount (150 g) of white bread corresponding to just over two slices, containing a nominal 50 g of available carbohydrate according to the manufacturer’s nutrition information panel (Nature’s Fresh, Goodman Fielder, Auckland, New Zealand). Participants were provided with a 250 mL glass of water. The bread was ingested within ten minutes, with baseline defined as the commencement of eating. Each participant remained sedentary in the two hours following carbohydrate ingestion on one morning and on the alternative morning cycled at very light intensity for 10 min after eating, commencing at either 15 or 45 min after baseline. Cycling was on seated ergocycles maintained at 40 rotations per minute on a setting of zero resistance.

### 2.5. Measurements

Anthropometric measurements of height and weight were recorded before the first morning test. Each test morning participants sat for a minimum of five minutes before two fasting capillary blood samples were taken. Capillary blood glucose values were then taken at 15, 30, 45, 60, 90, and 120 min after carbohydrate consumption, in line with published guidelines [31]. Capillary blood glucose over the testing period was measured on HemoCue 201+ systems (Radiometer, Copenhagen, Denmark). The coefficient of variation (CV) was 0.11%. The primary outcome was change in the postprandial blood glucose response between sedentary control and physical activity interventions. Incremental area under the blood glucose curve (iAUC) was calculated using the trapezoidal method [32].

### 2.6. Statistical Analysis

The sample size estimate was based on a power calculation at an alpha of 0.05 and a power of 0.80 to detect within group differences in outcome variables: A 0.5 mmol/L difference in capillary blood glucose at any time point or a 20% difference in iAUC. This estimate required 31 participants to complete both morning tests, however, we over-recruited to allow for dropouts. Data were analysed according to intention to treat. A mixed model was used to examine the difference in postprandial blood glucose response and included terms for both treatment and order. Blood glucose outcomes are presented comparing the physical activity intervention with the sedentary exposure. Comparisons among test conditions were made using a likelihood ratio (LR) chi-square statistic with 14 degrees of freedom comparing static differences between blood glucose at each time point and comparing iAUC among test conditions. Analyses were undertaken in Stata Version 13, (StataCorp, College Station, TX, USA) with the statistician blinded to the intervention type. 

## 3. Results

The characteristics of participants are shown in Table 1. There were no differences in the baseline measurements of participants randomised to physical activity starting at either 15 or 45 min post-baseline.

Physical activity starting 45 min after the commencement of eating coincided with the observed blood glucose peak. The postprandial blood glucose responses to physical activity are given in Figure 2.

There was no difference in any measured parameter of postprandial blood glucose when a matched bout of cycling began 15 min after carbohydrate ingestion when compared with the sedentary response. In the 15 min post-baseline test, the mean (SD) iAUC in the sedentary and active conditions were 140.5 (85) and 139.8 (91) respectively and were not different (*p* = 0.938). In the 45 min post-baseline test, the shape of the blood response curve differed when compared to the sedentary exposure (likelihood ratio chi-square = 31.47, *p* < 0.01). This difference in shape was driven by a reduction in blood glucose immediately following physical activity. The blood glucose concentration over the postprandial periods are given in Table 2.

The blood glucose difference at the 60 min time point in the group who started activity at 45 min post-baseline was 0.44 mmol/L (95% CI 0.14 to 0.74). The mean (SD) iAUC in the sedentary and active conditions were 128.2 (65) and 128.8 (62) and were not different (*p* = 0.842). When comparing between the 15 and 45 min groups, there was no significant difference in iAUC in the sedentary (*p* = 0.258) or active conditions (*p* = 0.483).

## 4. Discussion

Our results suggest that the timing of physical activity undertaken after carbohydrate ingestion influences the postprandial blood glucose response. Our study has matched the intensity and duration of activity, suggesting that the timing of activity is responsible for the observed differences. Activity at the blood glucose peak occurred when insulin secretion had likely plateaued [19] and the majority of carbohydrate digestion had occurred [33]. In contrast, activity undertaken before peak glucose may have reduced the insulin response, as found by Aadland and colleagues [16], and consequentially slowed the rate of glucose disposal. This is speculative and a limitation of our work as we did not measure postprandial insulin or the rate of glucose disposal. An acute bout of exercise has been found to increase glucose disposal rate in obese people and in people with type 2 diabetes, but not in lean participants [34]. Assessing the effect of light activity on the rate of glucose disposal in relation to the timing of commencement of that activity after eating would be an interesting area for future research.

Previous studies of physical activity and blood glucose control in normal glucose tolerant adults [16,18,19,20,21,22,29,30,35,36,37,38,39,40,41] did not use the timing of activity as a variable. In studies that have considered timing, there has been no within-study comparison between different timings within the postprandial period [26,27,28]. Furthermore, in studies where change in postprandial blood glucose response was not observed, even with activity of higher intensities the timing of activity may have been an unacknowledged determinant [29,36,39,42].

A difference in blood glucose concentration was found at the 60 min timepoint and in the shape of the glucose response curve when activity commenced 45 min after baseline, but there was no difference in iAUC between the active and sedentary conditions. By inspection of Figure 2 in the 45 min condition, it is apparent that there is a decrease in iAUC during activity, which is offset by an increase in iAUC occurring over the 90–120 min timeframe after activity. A rebound phenomenon whereby blood glucose concentration increases has been found previously under conditions of longer (45 min) activity at greater intensity (55% VO_2_ max), with those authors attributing it to ongoing carbohydrate absorption entering an environment of resting muscle [27]. However, in that study the blood glucose iAUC was less for the active condition compared with the sedentary condition. Our lack of difference is probably due to the short duration of the activity because blood glucose iAUC was lower after 20 min of slow walking compared with a sedentary control [26]. Therefore, it is clear that increasing the duration or intensity of the physical activity would have led to larger differences in postprandial blood glucose response. However, we specified very light intensity activity due to its low participant burden. Keeping the intensity of activity light is likely to be achievable for a wide range of individuals, including those with poor levels of fitness, with impairment, or people who are uncomfortable undertaking physical activity in public. It may be encouraging for some people to know that even small amounts of physical activity can make a difference. A limitation of our work is that it occurred for the duration of one meal only. Extending the current study to include several consecutive meals would have enabled us to test for a carry-over effect as previously reported [19,43,44,45]. Furthermore, consecutive bouts of activity throughout the day would serve to break up sedentary behaviours, an emerging independent factor in cardio metabolic risk [46].

Our results indicate the potential of physical activity to reduce blood glucose concentrations when they are high, including the postprandial period. Reduction of glycated haemoglobin in people with type 2 diabetes has been found both when targeting postprandial glycaemia with drugs [47] and with moderate physical activity undertaken for 40–50 min three times a week [48]. In a small study of two people, dampening of postprandial glycaemia and weight loss over one month was greater when walking for 30 min starting immediately after lunch and dinner compared with when an equivalent amount of exercise was started one hour after meals [49]. In another free-living crossover intervention, postprandial glycaemia was lower when 41 people with type 2 diabetes were advised to walk for 10 min after meals, starting 5 min after the finish of the meals, compared with when 30 min of activity was undertaken on a single daily occasion [50]. If people adopted just 10 min of post-meal activity after each of the three main meals of the day, this would make a contribution to fulfilling population-based activity recommendations with the added benefit of targeting postprandial glycaemia.

A limitation of the work is that two groups were studied. This design was adopted to suit the available resources and the demographics of the two groups were closely matched. Despite this limitation, a strength was that both groups had sample sizes larger than those of many other studies. Another limitation was the use of young, healthy adults. Activity could potentially be more beneficial for people with compromised glucose tolerance. However, it is encouraging to find an effect of very light activity on postprandial glucose dynamics even among healthy adults. A minimal intensity of activity was chosen on practical grounds, and we did not measure the effort. This is a limitation in the sense that we are unable to provide a numerical value for the power expended. Nevertheless, the study is reproducible if people use cycle ergometers set at zero resistance with a pedalling rate of 40 rotations per minute. Similarly, in another practically-oriented study, Lunde and colleagues did not measure power as the intervention simply required women to walk slowly [26]. Future research aimed at practically-achievable light activity interventions among people with conditions that could benefit from lowered glycaemia are warranted and urgently needed [51]. In our study, the potential for light activity to impact postprandial glycaemia was tested under controlled conditions after an overnight fast. Other factors that might impact the effect of activity on postprandial glycaemia could be time of day [52] and a second-meal effect, in which an earlier meal influences the glycaemic response to the following meal [53]. To explore effectiveness on a larger scale, the concept should be tested in the community to assess any effect on longer-term outcomes under usual living conditions, with options for people to choose a type and duration of activity to suit personal circumstances. In the meantime, physical activity is regarded as a cornerstone of diabetes management [54] and our results may have direct application for health care practitioners wanting to provide advice for control of postprandial blood glucose.

## 5. Conclusions

Our findings suggest that the timing of light physical activity shortly after eating affects the time-course of postprandial blood glucose. Activity initiated at the blood glucose peak may acutely lower blood glucose levels to a greater extent than the same amount of activity undertaken before the peak. These results support that activity, even for 10 min at very low intensity, may assist in the management of postprandial blood glucose if undertaken when blood glucose is high. Consumer acceptance might be high if the activity is easily achievable, while further work is necessary to consider these findings beyond the acute response and in people with impaired glucose tolerance.

## Figures and Tables

**Figure 1 nutrients-10-01743-f001:**
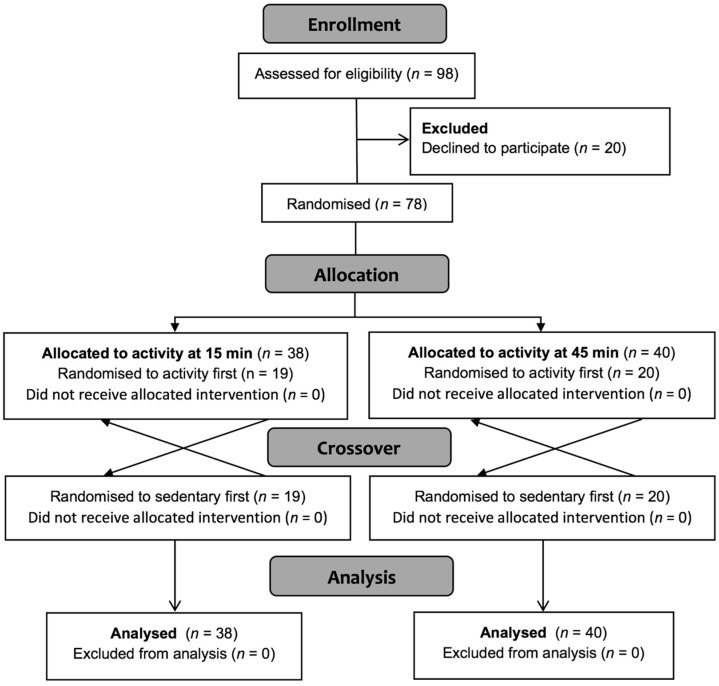
Consolidated Standards of Reporting Trials (CONSORT) diagram showing the flow of participants through the study.

**Figure 2 nutrients-10-01743-f002:**
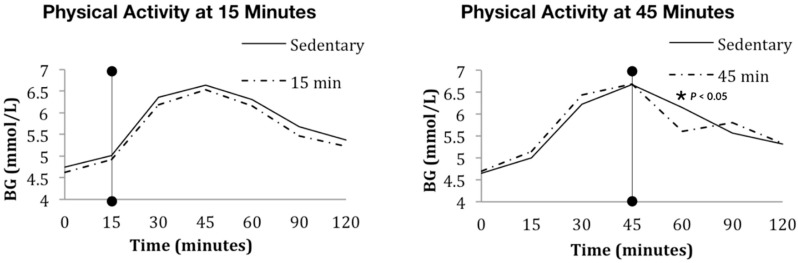
Blood glucose (BG) response to light cycling at 15 or 45 min after meal commencement. The vertical line with the filled circle ends represents the start of 10 min of cycling.

**Table 1 nutrients-10-01743-t001:** Participant characteristics.

Characteristic	Activity at 15 min (*n* = 38)	Activity at 45 min (*n* = 40)	*p*
Women/Men	32/6	30/10	0.915
BMI (kg/m^2^)	23.7 (4.07)	23.9 (3.63)	0.710
Age (year)	21.4 (1.35)	22.3 (5.16)	0.353
Fasting blood glucose (mmol/L)	4.65 (0.51)	4.70 (0.61)	0.384

Values are mean (SD). BMI is Body Mass Index.

**Table 2 nutrients-10-01743-t002:** Mean (SD) blood glucose concentration (mmol/L) during the sedentary and active arms in the groups assigned to 10 min of cycling starting either 15 or 45 min post-baseline.

Time (min)	Group Assigned to Activity Starting at Time = 15		Group Assigned to Activity Starting at Time = 45	
	Sedentary	Active	Sedentary	Active
0	4.7 (0.43)	4.6 (0.57)	4.7 (0.55)	4.7 (0.69)
15	5.0 (0.66)	4.9 (0.63)	5.1 (0.83)	5.2 (0.73)
30	6.3 (0.99)	6.2 (0.86)	6.3 (0.97)	6.5 (0.71)
45	6.6 (1.17)	6.5 (0.97)	6.7 (0.77)	6.7 (1.09)
60	6.3 (1.21)	6.2 (1.15)	6.1 (0.75)	5.6 (0.7) *
90	5.7 (0.84)	5.5 (0.82)	5.5 (0.71)	5.8 (0.76)
120	5.4 (0.76)	5.2 (0.73)	5.3 (0.84)	5.4 (0.78)

* Significantly different from the sedentary concentration.

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
