# Peer review of "The Timing of Activity after Eating Affects the Glycaemic Response of Healthy Adults: A Randomised Controlled Trial"

_nutrients, 2018, doi:10.3390/nu10111743_

Round 1
Reviewer 1 Report
The authors present a well-written report of a randomized, cross-over study evaluating the effect of light physical exercise on post-prandial glucose excursions. A few points to consider during revision:
abstract:
-line 22, clarify "commencement" here in the abstract.15 minutes from starting to eat or finishing eating?
introduction:
-line 44-45 and line 55-59: if possible, it would good to see a brief summary (intro or discussion) of the studies that have shown decreases in post-prandial glucose with post-meal exercise and the amount of lowering that was present. You talk about the level and timing of exercise (especially in discussion) but don't summarize any values of the lowering when it occurs to it can be compared to your observed 0.44mmol drop
-Line 55-59: how long after meal did these studies perform these exercise interventions.
Methods:
-were there any other exclusions or assessments to ensure young adults with normal glucose tolerance? OGTT?
-The authors should consider giving the full nutrition of the white bread. How many slices or weight was eaten by each participant? Was water allowed? Did you record the average time to eat the bread?
Results:
-could the authors consider providing a table or supplementary table of the means (SD) of the glucose level at each time-point measured. This will allow for future assessment and combing of data into meta-analyses, ect.
Discussion:
Line 154: it is probably less appropriate to draw conclusions about insulin response since you did not measure insulin. You should list a lack of insulin measurment as a limitation and future direction.
-could you discuss any clinical significant effect of the observed drop compared to other studies? Do you forsee this amount of drop cause any difference in long-term health outcomes (change in HbA1C%, etc)?
-You emphasize the likelihood of this intervention being more applicable to real-world but you should list limitations of such a statement since most people don't eat just bread within 10 minutes. Your findings give you evidence of the potential for a small intervention but further work is needed.
-Future directions should be expanded upon including increasing time of exercise, amount of meal, length of eating to make it more applicable to a real world setting.
Author Response
We would like to thank the reviewers for their insightful comments and hope that we have been able to address these.
abstract:
-line 22, clarify "commencement" here in the abstract.15 minutes from starting to eat or finishing eating?
Authors response: We have reworded as follows: “Bread was eaten within 10 min from baseline and no differences in postprandial blood glucose response were observed when cycling started 15 min after baseline compared with sedentary control.”
introduction:
-line 44-45 and line 55-59: if possible, it would good to see a brief summary (intro or discussion) of the studies that have shown decreases in post-prandial glucose with post-meal exercise and the amount of lowering that was present. You talk about the level and timing of exercise (especially in discussion) but don't summarize any values of the lowering when it occurs to it can be compared to your observed 0.44mmol drop
Author response: We have added the following to the Introduction: “Effects of low intensity activity carried out for short duration after eating have had variable effects on postprandial glucose excursions. Among 14 healthy women, slow walking for 15 min started immediately following a meal resulted in a 1.5 mmol/L reduction in blood glucose concentration at the end of the active period compared with a sedentary arm [22]. In contrast, blood glucose concentration was not different among 11 adults when eight minutes of moderate intensity cycling was undertaken immediately following eating compared with control [29]; and was higher by ~1 mmol/L 30 min after finishing 15 min bouts of cycling by six healthy volunteers compared with a control arm [30]. There may be a number of reasons for the discrepant findings including differences in participant demographics and study design, but of note the sample numbers were small. Given this heterogeneity in findings we were interested in exploring whether low intensity activity over a short duration could influence postprandial glycaemia of a larger group.”
-Line 55-59: how long after meal did these studies perform these exercise interventions.
Author response: That information is given in the preceding sentence, Lines 49-52
Methods:
-were there any other exclusions or assessments to ensure young adults with normal glucose tolerance? OGTT?
Author response: We have added the following to the Methods: “We recruited young adults without a self-reported diagnosis of dysglycaemia.”
-The authors should consider giving the full nutrition of the white bread. How many slices or weight was eaten by each participant? Was water allowed? Author response: We have added the following: “Each morning the participants consumed a weighed amount (150g) of white bread corresponding to just over two slices; containing a nominal 50g of available carbohydrate according to the manufacturer’s nutrition information panel (Nature’s Fresh, Goodman Fielder, Auckland, New Zealand). Participants were provided with a 250 mL glass of water.”
Did you record the average time to eat the bread?
Author response: No, we just ensured that participants had eaten it within 10 minutes.
Results:
-could the authors consider providing a table or supplementary table of the means (SD) of the glucose level at each time-point measured. This will allow for future assessment and combing of data into meta-analyses, ect.
Author response: We have provided Table 2.
Discussion:
Line 154: it is probably less appropriate to draw conclusions about insulin response since you did not measure insulin. You should list a lack of insulin measurment as a limitation and future direction.
Author response: Thank you, we have added the following: “This is speculative and a limitation as we did not measure postprandial insulin or rate of glucose disposal. An acute bout of exercise has been found to increase glucose disposal rate in obese people and in people with type 2 diabetes, but not in lean participants {Burstein, 1990 #119}. Assessing the effect of light activity on the rate of glucose disposal in relation to the timing of commencement of that activity after eating would be an interesting area for future research.”
-could you discuss any clinical significant effect of the observed drop compared to other studies? Do you forsee this amount of drop cause any difference in long-term health outcomes (change in HbA1C%, etc)?
Author response: We have added the following: “Reduction of glycated haemoglobin in people with type 2 diabetes has been found both when targeting postprandial glycaemia with drugs {Bastyr, 2000 #121} and with moderate physical activity undertaken for 40 – 50 min three times a week {Najafipour, 2017 #120}. In a small study of two people, dampening of postprandial glycaemia and weight loss over one month was greater when walking for 30 min starting immediately after lunch and dinner compared with when an equivalent amount of exercise was started one hour after meals {Hijikata, 2011 #123}. In another free-living crossover intervention, postprandial glycaemia was lower when 41 people with type 2 diabetes were advised to walk for 10 min after meals, starting 5 min after the finish of the meals, compared with when 30 min of activity was undertaken on a single daily occasion {Reynolds, 2016 #122}. If people adopted just 10 min of post-meal activity after each of the three main meals of the day, this would make a contribution to fulfilling population-based activity recommendations with the added benefit of targeting postprandial glycaemia.”
-You emphasize the likelihood of this intervention being more applicable to real-world but you should list limitations of such a statement since most people don't eat just bread within 10 minutes. Your findings give you evidence of the potential for a small intervention but further work is needed.
Author response: Thank you, we have added the following: “In our study the potential for light activity to impact postprandial glycaemia was tested under controlled conditions. To explore effectiveness on a larger scale, the concept should be tested in the community to assess any effect on longer-term outcomes under usual living conditions with options for people to choose a type and duration of activity to suit personal circumstances.”
-Future directions should be expanded upon including increasing time of exercise, amount of meal, length of eating to make it more applicable to a real world setting.
Author response: Please see response above.
Reviewer 2 Report
General comment
This study reports on a well-conducted randomised controlled trial (crossover but within parallel design) that assesses the timing of a light (10 min) bout of physical activity on postprandial glycaemia after ingestion of a 50g high GI meal (white bread) in healthy adult volunteers. Randomisation procedures were appropriate and effective and analyses were conducted by a statistician, blinded to the intervention allocation. The authors’ hypothesis and study design were well-considered and based on substantive knowledge as well as practical considerations. Although the findings are modest, they do suggest that light physical activity commenced 35 mins post-meal (peak of blood glucose response) may help with glycaemic control. I have only a few minor comments for the authors.
Discussion
179-181: The authors clearly highlight the study’s limitations and suggest that future research could examine consecutive meals. I think that this warrants further discussion and elaboration in relation to blood glucose responses to mixed meals and at different times of the day. My specific questions are:
1. Would the same timing of physical activity (i.e. 35 mins post meal) be appropriate following a mixed meal (e.g. with CHO, fat and protein). That is would the macronutrient and food composition of a meal affect the timing of the peak blood glucose response, and if so, how would you know when to prescribe a physical activity bout given people consume meals differing in size and composition in the real world. Further to this, would 35 mins be realistic following a large mixed meal and should gastric discomfort be a practical consideration?
2. The study was conducted in the morning and after fasting, at a time when glucose tolerance and insulin sensitivity is at its highest. Studies have shown that glucose tolerance and insulin sensitivity decrease throughout the day and into the evening. Would you expect time of day to play a role in the responsiveness of blood glucose to physical activity? Further to this, content and timing of previous meals have also been shown to affect glucose responses to subsequent meals. These points could be further discussed in relation to the rationale for examining consecutive meals in future research.
Author Response
We would like to thank the reviewers for their insightful comments and hope that we have been able to address these.
Would the same timing of physical activity (i.e. 35 mins post meal) be appropriate following a mixed meal (e.g. with CHO, fat and protein). That is would the macronutrient and food composition of a meal affect the timing of the peak blood glucose response, and if so, how would you know when to prescribe a physical activity bout given people consume meals differing in size and composition in the real world.
Author response: We had to ponder this when designing a walking intervention in people with type 2 diabetes in a free-living situation. We decided to ask people to start walking 5 min after finishing their breakfast, lunch and evening meals. The rationale for this was that it would take longer to finish a meal than it would to eat the bread we provided so that the glycaemic response would have been well underway by the time they started walking. The intervention was associated with a reduction in postprandial glycaemia compared with when activity was undertaken once daily in a single 30 min bout of walking. We also decided that it was best to keep the instruction as simple as possible. If we had asked participants to start timing the duration of their meals we were concerned that the participant burden may have reduced compliance. It may be that the timing of the start is not that critical, as long as it occurs in the proximity of the peak. In another study, waiting for one hour before starting to walk was found not to be as effective as walking immediately after eating. We have added the following to the Discussion: “In a small study of two people, dampening of postprandial glycaemia and weight loss over one month was greater when walking for 30 min starting immediately after lunch and dinner compared with when an equivalent amount of exercise was started one hour after meals {Hijikata, 2011 #123}. In another free-living crossover intervention, postprandial glycaemia was lower when 41 people with type 2 diabetes were advised to walk for 10 min after meals, starting 5 min after the finish of the meals, compared with when 30 min of activity was undertaken on a single daily occasion {Reynolds, 2016 #122}. If people adopted just 10 min of post-meal activity after each of the three main meals of the day, this would make a contribution to fulfilling population-based activity recommendations with the added benefit of targeting postprandial glycaemia.”
Further to this, would 35 mins be realistic following a large mixed meal and should gastric discomfort be a practical consideration?
Author response: As discussed above, waiting for 5 min to start walking after finishing meals was found to reduce postprandial glycaemia. The light intensity of the cycling activity was deliberately chosen so as not to risk discomfort. Similarly, in our participants in the Reynolds paper, walking at normal pace was recommended to avoid discomfort. In other studies of walking interventions after meals (eg: Nygaard, 2009, Lunde, 2012), discomfort has not been reported. We are not aware of problems of discomfort if people are undertaking walking or other activities at a low level of intensity.
The study was conducted in the morning and after fasting, at a time when glucose tolerance and insulin sensitivity is at its highest. Studies have shown that glucose tolerance and insulin sensitivity decrease throughout the day and into the evening. Would you expect time of day to play a role in the responsiveness of blood glucose to physical activity?
Authors response: Possibly, but the level of glycaemic response to meals will be multifactorial and in the study by Reynolds et al, the best outcome was found after the evening meal, possibly due to interrupting sedentary behaviour at this time of day.
Further to this, content and timing of previous meals have also been shown to affect glucose responses to subsequent meals. These points could be further discussed in relation to the rationale for examining consecutive meals in future research.
Author response: We agree, but from a practical viewpoint there’s little that can be done about that and it’s probably better that the activity is undertaken regardless. We have added the following: “In our study the potential for light activity to impact postprandial glycaemia was tested under controlled conditions after an overnight fast. Other factors that might impact the effect of activity on postprandial glycaemia could be time of day {Leung, 2017 #125} and a second-meal effect, in which an earlier meal influences the glycaemic response to the following meal {Jovanovic, 2009 #124}. To explore effectiveness on a larger scale, the concept should be tested in the community to assess any effect on longer-term outcomes under usual living conditions with options for people to choose a type and duration of activity to suit personal circumstances.”